# Evaluation of tilt control for wind-turbine arrays in the atmospheric boundary layer

Carlo Cossu

Laboratoire d'Hydrodynamique Énergetique et Environnement Atmosphèrique (LHEEA)
CNRS - Centrale Nantes, 1 rue de la Noë 44300 Nantes, France

**Correspondence:** Carlo Cossu (carlo.cossu@ec-nantes.fr)

**Abstract.** Wake redirection is a promising approach designed to mitigate turbine-wake interactions which have a negative impact on the performance and lifetime of wind farms. It has recently been found that substantial power gains can be obtained by tilting the rotors of spanwise-periodic wind-turbine arrays. Rotor tilt is associated to the generation of coherent streamwise vortices which deflect wakes towards the ground and, by exploiting the vertical wind shear, replace them with higher-momentum fluid (high-speed streaks). The objective of this work is to evaluate power gains that can be obtained by tilting rotors in spanwise-periodic wind-turbine arrays immersed in the atmospheric boundary layer and, in particular, to analyze the influence of the rotor size on power gains in the case where the turbines emerge from the atmospheric surface layer. We show that, for the case of wind-aligned arrays, large power gains can be obtained for positive tilt angles of the order of $30^o$. Power gains are substantially enhanced by operating tilted-rotor turbines at thrust coefficients higher than in the reference configuration. These power gains initially increase with the rotor size reaching a maximum for rotor diameters of the order of 3.6 boundary layer momentum thicknesses (for the considered cases) and decrease for larger sizes. Maximum power gains are obtained for wind-turbine spanwise spacings which are very similar to those of large-scale and very large scale streaky motions which are naturally amplified in turbulent boundary layers. These results are all congruent with the findings of previous investigations of passive control of canonical boundary layers for drag-reduction applications where high-speed streaks replaced wakes of spanwise-periodic rows of wall-mounted roughness elements.

## 1   Introduction

An unavoidable byproduct of wind-turbine operation is the generation of wakes containing the low-speed, highly turbulent fluid from which mechanical power has been extracted. As in wind farms the streamwise distance between wind turbines is typically much shorter than the distance required for the wake to diffuse and recover the incoming wind speed, wind-farm interior turbines experience large reductions of their power production and significant unsteady loads when they are shadowed by wakes of upstream turbines (see e.g. Stevens and Meneveau, 2017; Porté-Agel et al., 2019, for recent reviews). Among the significant number of techniques proposed to minimize the negative effects of turbine-wake interactions, the approach where wakes are deflected away from downstream turbines by yawing upstream rotors has recently attracted much interest.

It has indeed been shown that power gains in downstream turbines associated to wake deflections can exceed power losses experienced by yawed turbines (Dahlberg and Medici, 2003; Jiménez et al., 2010).

In addition to yaw control, where wakes are deflected horizontally, it has been more recently proposed to deflect wakes in the vertical direction by acting on the rotor-tilt angle (see e.g. Guntur et al., 2012; Fleming et al., 2014, 2015; Annoni et al., 2017; Bay et al., 2019; Cossu, 2020; Scott et al., 2020; Nanos et al., 2020). Best performances are obtained for positive tilt angles, where the wake is deflected towards the ground (or the sea surface). This type of control, with positive tilt angles, can not be implemented in most of current-generation wind turbines which have upwind-facing rotors (for which positive tilt would lead to blade-tower hits) and which have been designed with very limited tilt capabilities. Positive tilt is, however, well suited for turbines with downwind-oriented rotor whose interest is being currently reconsidered because of their favorable distributions of blade bending loads, which are critical in the design of next-generation light and flexible blades, and their resilience in off-grid/extreme wind conditions (Loth et al., 2017; Kiyoki et al., 2017). In the case of offshore applications, recent studies also show the possibility of obtaining positive rotor tilt angles for upwind-oriented turbines by tilting the whole floating turbine with differential ballast control (Nanos et al., 2020). However, in this case, unsteady tilt effects related to the whole turbine pitching motion could also be important. These unsteady effects are outside the scope of this study and, in the following we limit our discussion to the case where the rotor tilt angle (with respect to the vertical axis) is steady.

Global power gains are achievable also for tilt control despite the power losses experienced by tilted-rotors upstream turbines (Fleming et al., 2015; Annoni et al., 2017; Bay et al., 2019; Cossu, 2020; Scott et al., 2020; Nanos et al., 2020). In a companion paper (Cossu, 2020, referred to as C2020 in the following), we have recently shown that by tilting upstream rotors of spanwise-periodic turbine rows, turbine wakes (low-speed fluid) can be replaced with high-speed streaks, i.e. streamwise-elongated regions where the wind speed exceeds the mean, similarly to what observed in passive control approaches designed for drag-reduction applications (Fransson et al., 2006; Pujals et al., 2010b). An important aspect of the tilt-control approach is that it exploits the beneficial effect of the vertical wind shear, which is related to the lift-up effect by which streaks are amplified (Moffatt, 1967; Landahl, 1980; Schmid and Henningson, 2001) with a non-modal amplification mechanism (Böberg and Brosa, 1988; Butler and Farrell, 1992; Gustavsson, 1991; Trefethen et al., 1993; Cossu et al., 2009; Cossu and Hwang, 2017). Such a beneficial effect is not exploited by yaw control where the wakes are deflected in the horizontal plane and therefore tilt control can potentially lead to power gains larger than those associated to yaw control.

In C2020 it was found that significant improvements of the global power production can be obtained by operating tilted-rotor turbines at higher induction (higher thrust coefficient) to compensate for the reduction of the normal velocity component. It was also shown that those power gains could be further improved when considering higher values of the $D/H$ ratio of the rotor diameter $D$ to the boundary layer height $H$, at least for $D/H < 0.36$, in accordance with theoretical predictions of streak amplifications in turbulent wall-bounded shear flows (Cossu et al., 2009; Pujals et al., 2009; Hwang and Cossu, 2010; Cossu and Hwang, 2017) and with previous experimental results where streaks were forced with roughness elements instead of tilted-rotor wind turbines (White, 2002; Fransson et al., 2004; Pujals et al., 2010a). Global power gains of the order of 40% were attained for the largest considered ratio $D/H = 0.36$.

The results of C2020, nevertheless, call for confirmation in the high-$D/H$ range because they do not necessarily extrapolate to atmospheric boundary layers since they were obtained in a pressure-driven boundary layer (PBL) where the effects of Coriolis acceleration and capping inversion were neglected, and furthermore with rotors taller than the logarithmic layer region mimicking the atmospheric surface layer. The scope of the present investigation is therefore to evaluate the level of power gains that can be obtained by tilting upstream rotors in wind-turbine arrays immersed in atmospheric boundary layers (ABL) where the effects of Coriolis acceleration and capping inversion are fully taken into account.

The questions in which we are interested are the following: What are the typical power gains that can be obtained by tilt-control in the ABL? How do they compare to those found in the PBL? How do these power gains depend on the relative rotor size, especially in the case of large rotors? To answer these questions, we use large-eddy simulations (LES) to simulate neutral atmospheric boundary layers with capping inversions in the presence of wind-turbine arrays in wind-aligned configurations which are the worst case for turbine-wake interactions. The turbines are modeled with the actuator-disk method and are assumed to operate at constant thrust coefficient in order to obtain generic results which do not depend of the specificity of particular control laws chosen for turbines operation. The effect of tilt angle, induction factor and rotor size on the global power production will be studied with respect to reference configurations where the turbines are operated in standard mode.

The paper is organized as follows: the problem formulation is introduced in §2, results are presented in §3 and summarized and further discussed in §4. Additional details on used numerical methods are provided in Appendix A and some additional results in Appendix B.

## 2   Problem formulation

We consider the flow developing on wind-turbine arrays immersed in the atmospheric boundary layer. The flow is computed by means of large-eddy simulations implemented in SOWFA, the NREL's Simulator for On/Offshore Wind Farm Applications which solves the filtered Navier-Stokes equations (see Churchfield et al., 2012, and Appendix A for additional details). The horizontal component of the Coriolis acceleration is included in the equations, compressibility effects are accounted for by means of the Boussinesq approximation and the Smagorinsky (1963) model is used to approximate subgrid-scale stresses. It is assumed that Monin-Obukhov similarity theory for turbulent boundary layers above rough surfaces applies near the ground by implementing appropriate stress boundary conditions as in Schumann (1975) while slip boundary conditions are enforced at the upper plane of the solution domain. Periodic boundary conditions are applied in the spanwise direction with $L_y$ periodicity (the spanwise extension of the domain).

Preliminary 'precursor' simulations of the atmospheric boundary layer in the absence of wind turbines are run with periodic boundary conditions enforced also in the streamwise direction (with $L_x$ streamwise periodicity). The precursor simulations are used to generate realistic inflow wind conditions (Keating et al., 2004; Tabor and Baba-Ahmadi, 2010; Churchfield et al., 2012) for the simulations with wind turbines by storing the temporal evolution of flow variables on a vertical plane which is then used as inflow boundary condition for the simulations in the presence of wind turbines.

The actuator disk model is used to approximate the forces exerted by wind turbines on the fluid and the power produced by the turbines. This model has been shown to provide reliable results for the characteristics of turbines wakes except in the wake formation region (Wu and Porté-Agel, 2011). To obtain general results, not depending on the specific control law assumed for the considered turbines, the total force exerted by each turbine on the fluid is assumed to be of the form $\mathbf{F} = -C_T' \rho u_n^2 A \mathbf{e}_n / 2$, as done by e.g. Calaf et al. (2010), Goit and Meyers (2015) and Munters and Meyers (2017), where $C_T'$ is the disk-based thrust coefficient, $\mathbf{e}_n$ is the unit vector normal to the rotor disk, $u_n$ is the wind velocity component normal to the rotor averaged over the disk surface of area $A = \pi D^2 / 4$ and $D$ is the rotor diameter. The force is assumed to be uniformly distributed on the rotor and wake rotation effects are neglected because their modeling would reintroduce a dependence on specific turbine design and control. Turbines are assumed to (always) operate in Region II at constant $C_T'$.

The power produced by each turbine is modeled, for all the results shown, as $P = C_P' \rho u_n^3 A / 2$ where $C_P' = \chi C_T'$ with the coefficient $\chi = 0.9$ accounting for wing-tip power losses and LES (coarse) grid effects (Martínez-Tossas et al., 2016; Shapiro et al., 2019). Power *gains* that will be computed in the following, however, being ratios of powers computed with the same $\chi$, do not depend on the specific value chosen for $\chi$. The value $C_T' = 2$ corresponds to the optimal Betz value maximizing the power output of an isolated ideal turbine (Burton et al., 2001; Munters and Meyers, 2017).

We will consider the effect of rotor tilt in arrays composed of three rows of wind turbines aligned with the mean wind and spaced by $\lambda_x = 7D$ in the streamwise direction as in Annoni et al. (2017), who considered a single column of three aligned turbines immersed in the ABL, and C2020 who studied three spanwise-periodic arrays immersed in the pressure boundary layer (PBL). The arrays are periodic in the spanwise direction with spanwise turbine spacing $\lambda_y = 4D$ as in C2020, a value inspired by the spanwise spacing used for cylindrical roughness elements of width/diameter $D$ used as 'streak generators' in previous boundary layer control studies as those of Fransson et al. (2004, 2005, 2006), Hollands and Cossu (2009), Pujals et al. (2010b) and Pujals et al. (2010a). Tilt control will be applied to the two upwind rows of turbines by changing their rotor-tilt angle $\varphi$ and thrust coefficient $C_T'$ while turbines in the third, downwind, row are left at reference conditions.

## 3 Results

### 3.1 Precursor simulations of the ABL

In the precursor simulations we consider neutral atmospheric boundary layers (ABL) at latitude $41^o N$, whose thickness is kept almost steady by the presence of a capping inversion centered at $z = H$ and of thickness $\Delta z_{CI}$, inside which there is a positive vertical potential temperature gradient $(d\theta/dz)_{CI}$. The potential temperature $\theta_{neut}$ below the capping inversion is constant ($300K$), while a constant positive vertical gradient $(d\theta/dz)_G = 0.03K/m$ is enforced in the geostrophic region above. The flow is driven by a pressure gradient enforcing mean westerly winds of 8m/s at $z = 100m$. Three boundary layers are considered with $H = 750m$, $H = 500m$ and $H = 350m$ respectively, as detailed in Table 1. The precursor simulations are run up to $t_1 = 20000s$ to extinguish initial transients and obtain a well-developed ABL. The driving mean pressure gradient and the flow variables on the west boundary ($x = 0$) are then stored from $t_1$ to $t_2 = 30000s$, i.e. for more than two and a half hours, to be used as driving term and inflow boundary conditions in the simulations with wind turbines.

The flow is simulated in two sets of domains, the first extending 3km x 3km in the streamwise ($x$, west-east) and spanwise ($y$, north-south) directions respectively and the second of 6km x 3km extension all discretized with cells of $15m$ size in the horizontal directions and size ranging from $7m$ (near the ground) to $21m$ (in the freestream above the capping inversion) in the vertical direction.

The mean-wind profiles obtained for the three selected $H$ in the 3km x 3km domains are nearly indistinguishable up to $z \approx 250m$, where they are well fitted by the logarithmic law (see Fig. 1). In this region the wind direction is essentially directed along the $x$ axis (from west to east) with a small $y$ (north-south) component which is increasingly southerly for increasing heights. Identical results are found in the 6km x 3km domains (not shown).

## 3.2 Effect of tilt angle and $C'_T$ on power gains

We first consider the effect of tilt on wind turbines with rotor diameter $D = 126m$ and hub height $z_h = 89m$ (the same dimensions as the NREL 5-MW turbine model defined by Jonkman et al., 2009) in the 3km x 3km domain which accommodates three rows of six turbines each (see Fig. 3). The simulations with wind turbines are performed in the same numerical domain, with the same grid and parameters of the precursor simulations for $10000s$ using the driving mean pressure gradient and the west-plane ($x = 0$) inflow boundary conditions recorded in the precursor simulation. Statistics are accumulated for the last $6000s$ of the simulation.

First, a reference case is simulated in the ABL with $H = 750m$ where all turbine rotors have the usual $-5^o$ tilt angle used to avoid blade-tower collisions for standard upwind-aligned rotors (Jonkman et al., 2009) and are operated at constant $C'_T = 1.5$ which is consistent with those observed in real wind farms (Wu and Porté-Agel, 2015; Stevens et al., 2015; Munters, 2017). Then, a set of controlled cases are considered where the rotors of the two most upwind turbine rows are tilted all by the same angle $\varphi$ (as in Annoni et al., 2017, and C2020).

The ratios of the global power produced in the controlled case to the global power produced in the reference case are reported in Fig. 2$a$ for the selected tilt angles $\varphi = 20^o$, $30^o$ and $40^o$. From Fig. 2$a$ it is seen that, for the present case where all turbines are operated at the same $C'_T = 1.5$, best power gains above 15% are obtained for $\varphi = 30^0$. These gains are of the same order of those found by Annoni et al. (2017) for three aligned turbines in the ABL (using the specific NREL5 load distribution and control law) and those found by C2020 in the pressure boundary layer (PBL) for the same array configuration and operation.

Following C2020, the simulations are repeated operating tilted-rotor turbines at higher $C'_T$. To clearly isolate the effect of operating tilted turbines at higher $C'_T$ on tilt control, results will be shown for the large value $C'_T = 3$ (the same value considered by C2020) that is near the maximum of the range attainable by real turbines (see e.g. Goit and Meyers, 2015; Munters and Meyers, 2017; Cossu, 2021). Quantitatively intermediate whilst qualitatively similar results, are obtained for intermediate values of $C'_T$, as shown in Appendix B. From Fig. 2$a$ we see that for $C'_T = 3$ power gains have almost doubled, attaining values above 30% for $\varphi = 30^o$, similarly to what previously found in the PBL.

From Fig. 2$b$, where relative powers extracted by each row of turbines are reported, it can be seen that the effect of tilt is to moderately increase the power extracted by the middle row of turbines and greatly increase the power extracted by the most downwind row. The effect of tilt, indeed, is to create a pair of counter-rotating streamwise vortices (Dahlberg and Medici, 2003;

Howland et al., 2016; Bastankhah and Porté-Agel, 2016) which are horizontally-staked (see Fig. 4$b$) and are associated with a strong (vertical) downwash in the wake of the tilted rotors (see Fig. 4$b$ and the blue regions of negative vertical velocity clearly discernible in Fig. 3$d$). The downwash induced by the first and second row of turbines is almost additive (see Fig. 3$d$). From panels $a$ and $b$ of Fig. 3 and from Fig. 4 it can be seen that the forced downwash is strong enough to expel the low-speed region (the wake) first towards the ground and then on the sides, and replace it with high-speed streaks by exploiting the positive wind shear (Fleming et al., 2015; Annoni et al., 2017; Cossu, 2020).

These results, obtained in the ABL, are consistent with those found in the pressure boundary layer (PBL), as could be expected because for the considered $D = 126m$ turbines and the $H = 750m$ ABL ($D/H = 0.126$) the rotors are mostly immersed in the atmospheric surface layer, where the velocity profile is logarithmic (see Fig. 1$a$) and which is well approximated by the inner region of the PBL. In the considered case, the slight wind veering with height has for effect only a slight bending of the wake region and of the high-speed region in the cross-stream plane (see Fig. 4) with no apparent significant influence on the coherent lift-up process by which streaks are amplified (Cossu et al., 2009; Pujals et al., 2009; Cossu and Hwang, 2017; Cossu, 2020).

### 3.3 Influence of the relative rotor size on power gains

In C2020, for the case of the pressure boundary layer (PBL), increasing power gains were found for increasing $D/H$ ratios of the rotor diameter to the boundary layer thickness, at least up to $D/H = 0.36$. As already mentioned, however, for the largest values of $D/H$ considered in C2020 the results obtained in the PBL are not necessarily a good approximation of those that one would find in the ABL because rotors emerged from the logarithmic region where PBL and ABL mean wind profiles are similar. Furthermore, in the PBL considered in C2020 a slip boundary condition at $z = H$ was enforced, which could also have artificially affected streak growths for the largest considered $D/H$ values. We therefore investigate if the results found in the PBL hold also in the ABL. In the present ABL setting, where $H$ is given by the capping inversion height, we will also be able to access values of $D/H$ larger than in C2020.

We begin by considering the effect of reducing $H$ from $H = 750m$ to $H = 500m$ and then to $H = 350m$ for the same array of turbines with $D = 126m$ considered in §3.2 (where tilted-rotor turbines are operated at $C_T' = 3$ while the others are operated at the reference $C_T' = 1.5$). Reducing $H$ at constant $D$ results in increasing the $D/H$ ratio. From Fig. 5$a$, where the results are reported, it is seen that for all considered $H$, the maximum power gains are obtained for $\varphi = 30^o$ and that for this tilt angle they do initially increase with $D/H$ but then reach a maximum and decrease for the largest considered $D/H$ values.

To better explore the high-$D/H$ regime, additional simulations are performed with larger turbines keeping constant the relative spacing of 4D and 7D in the spanwise and streamwise directions respectively and the operation mode ($C_T' = 3$ for tilted rotors, $C_T' = 1.5$ for the others). First, an array of turbines with $D = 180m$ and $z_h = 115m$ (inspired by the DTU-10 MW model) is considered in the same 3km x 3km domains already used (three rows with four turbines each are accommodated in the domain). Then, arrays of even larger turbines with $D = 250m$, $z_h = 150m$ and $D = 360m$, $z_h = 230m$ are accommodated in 6km x 3km domains (three rows with 3 and 2 turbines respectively in each row). For all considered D-H combinations the effect of tilt on the flow fields is similar to that shown in Fig. 3, with wind-veer induced deviations to the south of the

forced high-speed streaks becoming relevant for the $D = 360m$ turbines in the $H = 500m$ ABL, corresponding to the largest considered ratio $D/H = 0.72$.

Also for the $D = 180m$ turbines (as shown in Fig. 5b) and for the $D = 250m$ and $D = 360m$ turbines (not shown) the best power gains are obtained for $\varphi = 30^o$, a value which is therefore robust to the change of $D$ and $H$. From Fig. 5b it is seen that also for $D = 180m$ turbines, the best power gains are obtained in the $H = 500m$ ABL ($D/H = 0.36$) and not in the $H = 350m$ one corresponding to the largest value $D/H = 0.51$. Similar results are found for $D = 250m$ and $D = 360m$ turbines.

To better appreciate the influence of $H$ and $D$ on power gains, the $P/P_{ref}$ values obtained for $\varphi = 30^o$ are reported as a function of $D/H$ in Fig. 6 for all the considered $D$-$H$ combinations and are compared to those obtained in the PBL (reproduced from data reported in C2020). From Fig. 6 it is seen that, for all the considered atmospheric boundary layers, power gains initially increase with turbine diameter (and therefore $D/H$) before reaching a maximum value ($D/H \approx 0.24-0.51$ depending on the considered $H$) and eventually decrease when $D$ is further increased.

The saturation of power gains at sufficiently large $D/H$ can not be attributed to lateral wake/streaks displacements associated to strong wind-veer effects because, for all considered ABLs, saturation is reached for the $D = 180$ turbines, which remain in the region where mean velocity profiles are identical for the three considered $H$ and wind veer is negligible (see Fig. 1). The saturation can probably be associated to the existence of an optimal spanwise wavelength where the amplification of the streamwise streaks induced by the streamwise vortices generated by the tilted rotors is maximum, as theoretical models of optimal energy amplifications predict (Pujals et al., 2009; Cossu et al., 2009; Hwang and Cossu, 2010) and as is observed in experiments of forced streaks amplifications in turbulent boundary layers (Pujals et al., 2010b). Following this line of thought, the maximum of $P/P_{ref}$, obtained for $D/H \approx 0.24-0.51$, i.e. for a spanwise wavelength of $\lambda_y = 4D \approx 1-2.5H$, would therefore correspond to the spanwise wavelength of maximum amplification of the streamwise streaks leading to the largest mean velocity increase in correspondence to downstream rotors.

From Fig. 6 it is also seen that $P/P_{ref}$ data are quite dispersed when plotted as a function of $D/H$ and, most importantly, maximum power gains are attained at values of $D/H$ which strongly vary from one boundary layer to another. This could have been expected, since the considered turbulent boundary layers are not self-similar. Previous investigations (see e.g. Corbett and Bottaro, 2000; Cossu et al., 2009) had, however, shown that the most relevant length-scale selecting optimal spanwise wavelengths $\lambda_y$ associated to maximum streaks amplification in non-self-similar boundary layers is the boundary layer momentum thickness $\delta_2 = \int_0^\infty [1 - U(z)/U_\infty][U(z)/U_\infty]dz$, where $U_\infty$ is the streamwise velocity in the freestream above the boundary layer. We therefore replot in Fig. 7 the $P/P_{ref}$ data as a function of $D/\delta_2$. From Fig. 7 it is seen that this new rescaling (on $\delta_2$ instead of $H$) greatly reduces the data dispersion and that maximum gains are obtained for $D/\delta_2 \approx 3.6$. This enhanced scaling based on $\delta_2$ provides further evidence that, also in the ABL, streaks amplification is the main driving mechanism of the observed power gains obtained by tilting rotors.

## 4 Conclusions

In this study we have evaluated the gains of global extracted wind power that can be obtained by tilting rotors in wind-turbine arrays immersed in neutral atmospheric boundary layers with capping inversions. In particular, we have considered spanwise-periodic arrays with three turbine rows where rotors of the two upwind rows are all tilted by the same angle $\varphi$ while rotors of the downwind row are left in reference conditions.

It is found that significant power gains can be obtained. For all the considered combinations of turbine rotors diameters ($D = 126m$, $D = 180m$, $D = 250m$ and $D = 360m$), boundary layer heights ($H = 750m$, $H = 500m$ and $H = 350m$), and $C'_T$ of operation, the best power gains are obtained for tilt angles $\varphi \approx 30^o$. Power gains are improved when tilted turbines are operated at induction rates higher than the reference one (corresponding to $C'_T = 1.5$). Results have been presented for the largest considered value $C'_T = 3$, where power gains are highly enhanced, but substantial power gains are already reached at intermediate $C'_T$ values, as shown by additional results reported in Appendix B.

The influence of rotor size on power gains has also been investigated for ratios $D/H$ up to 0.72, larger than those previously considered by Cossu (2020) in the PBL. Results show that power gains do initially increase with rotor size (starting from small $D/H$ values) but they eventually saturate and decrease for sufficiently large ratios $D/H$.

Following the rationale of Cossu (2020), relating tilt-induced power gains to the amplification of large-scale streaks forced by tilted rotors, the existence of an optimal rotor diameter maximizing tilt-induced power gains has been connected to the existence of an optimal spanwise wavelength maximizing streak amplifications. Credit to this interpretation is given by the fact that the range of optimal spanwise turbine spacings $\lambda_y = 4D \approx 1 - 2.5H$ corresponds to the spacing of large-scale streaks associated to large-scale and very-large-scale streaky motions (LSM and VLSM) in neutral turbulent boundary layers (see e.g. Hutchins et al., 2012; Shah and Bou-Zeid, 2014; Fang and Porté-Agel, 2015) whose natural amplification is based on the same mechanism (Pujals et al., 2009; Cossu et al., 2009; Hwang and Cossu, 2010; Cossu and Hwang, 2017). Further credit to this interpretation is given by the fact that power gain data obtained for all the considered $D$-$H$ turbine-ABL combinations do scale better on $D/\delta_2$ than on $D/H$ ratios, where $\delta_2$ is the boundary layer (streamwise) momentum thickness, as do streak-energy amplifications in non-self-similar boundary layers (Corbett and Bottaro, 2000; Cossu et al., 2009).

Additional final evidence is, nonetheless, needed to confirm the relation between streaks amplification and tilt-induced power gains in the ABL by comparing streamwise vortices forced by tilted rotors to the optimal perturbations leading to maximum large-scale streaks amplification in the ABL and in particular their optimal spanwise spacing. Optimal perturbations and energy amplifications in the ABL are, however, currently unknown (they have, actually, been computed in the Ekman boundary layer by Foster, 1997, but not for ABL profiles with logarithmic profiles in the surface layer). This is the subject of current intensive research effort.

It is also important to emphasize that a non-negligible part of the obtained power gains is associated to the operation of tilted-rotor turbines at higher induction rates. In the present study and in the previous related one of Cossu (2020) only a few basic combinations of tilt angles $\varphi$ and (steady) thrust coefficients $C'_T$ have been considered for tilted-rotor turbines. It is believed that there is large room for further power-gain improvements that would come from the optimization of the $\varphi - C'_T$

distribution among controlled turbines. Further work is also needed to evaluate power gains attainable for typical wind roses instead of the single wind-aligned case considered here. In this context, it is important to remark that for tilt control one not only steers the (low-speed) wake away from the downstream rotor but also targets at it the forced high-speed streak. In a case where the arrays are not aligned with the mean wind, therefore, one will probably have to decide if it is more convenient to combine tilt control with yaw control to target the high speed streak at the downstream rotor or to switch-off the tilt control. Combined tilt-yaw control might also be necessary in situations of increased wind veer.

One limitation of the present study resides in the used highly idealized actuator disk model for the turbines which are assumed to operate at constant $C_T'$ and where aerodynamic forces are assumed to be uniformly distributed and purely normal to the rotor disk, therefore neglecting wake rotation effects. Whilst such a modeling was required to compare turbines of different diameters and avoid a dependence on specific controllers, additional investigations, based on more realistic turbine models and control laws, are needed to assess the level of practically attainable power gains and their dependence of the chosen control laws. More realistic turbine models are especially needed in high-$D/H$ regimes where rotors are large enough to directly interact with the capping inversion and the geostrophic free-stream. Such regimes would not only be encountered for future-generation very large turbines but also in current-generation wind turbines immersed in stable ABLs where boundary layer heights $H$ are small. In this case it is known that wake rotation effects are important (see e.g. Englberger et al., 2020) and wind veer and Coriolis acceleration are likely to strongly influence the streak amplification process, especially trough a modification of the trajectory of the pair of counter-rotating streamwise vortices generated by the tilt. Moreover, as our results are based on large-eddy simulations, they might be sensitive to the used subgrid-scale model, so that experimental verification and further analysis with different subgrid stress models would be welcome especially in what concerns the turbulent diffusion of the tilt-induced quasi-streamwise vortices which play an important role in generating the high-speed streaks relevant to tilt-control.

Finally, it is important to note that this study has dealt only with the fluid dynamics of tilt-control without addressing issues such as the level of structural loads that turbines would experience when tilted by angles as large as $30^o$, possibly with increased values of $C_T'$. A first reason for this choice is that it is necessary to know the range of parameters where the tilt control aerodynamics is most efficient before addressing the question of its practical implementation. An additional reason is that it might be questionable to draw conclusions from computations of such kind of structural loads on the available models of existing turbines which have not been designed to be tilted by large angles, or even with the required positive tilt angles as they have upwind-facing rotors. It is therefore important that structural loads experienced when the rotor is significantly tilted, especially when turbines are operated at higher $C_T'$, are computed in future studies, including new tilt-optimized designs, of future-generation large-scale turbines probably with downwind-oriented rotors which would enable operation at positive tilt angles such as those discussed by Loth et al. (2017). A practical implementation of tilt control in future-generation wind farms will be possible, indeed, only if an acceptable compromise can be found between operational demands of tilt-control (tilt angles, rotor loading) and structural and cost constraints.

## Appendix A: Methods

The filtered Navier-Stokes equations, including the effect of Coriolis acceleration, with Boussinesq fluid model and the Smagorinsky (1963) model of turbulent subgrid scale motions are solved with SOWFA (Churchfield et al., 2012), an open-source code based on the OpenFOAM finite-volumes code (OpenCFD, 2011). The PIMPLE scheme is used for time advancement. Schumann (1975) stress boundary conditions are enforced at the near-ground horizontal boundary with a characteristic roughness length $z_0 = 0.001$ typical of offshore conditions.

Numerical simulations have been performed in numerical domains of streamwise length $L_x$ (3km and 6km) and width $L_y$ (3km). Three domains heights $L_z$ have been considered as detailed in Table 1. Totally, six numerical domains have therefore been used in the simulations. All the domains are discretized with cells of constant $15m$ size in the horizontal directions and height-increasing size ranging from the $7m$ to $21m$ in the vertical direction. The solutions are advanced with $\Delta t = 0.8s$ time steps which keeps reasonable the amount of data stored in precursor simulations.

In order to enforce the turbine response to depend only on $C_T'$ and not on the turbine controller, for the sake of simplifying the interpretation of the results, an in-house ADMC model has been derived from the original actuator disk (ADM) turbine model implemented in SOWFA (Martínez-Tossas and Leonardi, 2013) by distributing the body force uniformly in the radial direction while keeping the same discretization points on the disk and setting its magnitude to $\mathbf{F} = -C_T' \rho u_n^2 A \mathbf{e}_n / 2$ (Calaf et al., 2010; Goit and Meyers, 2015; Munters and Meyers, 2017) and setting to zero body force components parallel to the rotor plane, thus setting to zero the wake rotation. A Gaussian projection of discretized body forces with a smoothing parameter $\varepsilon = 20m$ is also used to avoid numerical instabilities (Sørensen and Shen, 2002; Martínez-Tossas and Leonardi, 2013).

## Appendix B: Further results on the influence of $C_T'$

In §3.2 it has been shown that substantial power gain enhancements can be obtained by combining tilt control with high-induction operation of tilted turbines. Results were shown for the case where the turbines are operated at $C_T' = 3$, as in C2020. For the case of the NREL-5MW (D=126m) turbine model, recent studies show that $C_T' = 3$ operation is near the upper limit of what can be realized in practice by either increasing the tip speed ratio or/and the blades chord length (Goit and Meyers, 2015; Munters, 2017) or by acting on the rotor-collective blade-pitch angle (Cossu, 2021). However, as detailed analyses of static and dynamic structural loads are not yet available for the case of tilted turbines operated at such high $C_T'$ values, it is not yet clear that the $C_T' = 3$ case is the best one when turbine cost and life-expectancy are taken into account in the optimization process. In this Appendix we therefore report some further results showing which level of power gains that an be obtained at $C_T'$ values lower than 3.

In Fig. B1$a$ the influence of $C_T'$ on the power produced by each of the three turbine rows is shown for the case of D=126m turbines in the H=750m ABL with $\varphi = 30^o$ tilt control; from this figure it is seen that by increasing $C_T'$, the power of all the three rows of turbines is increased but almost saturating when approaching $C_T' = 3$. In Fig. B1$b$ the combined effect of changing $C_T'$ and the ABL depth $H$ on the total produced power is analyzed for the case of D=180m turbines with $\varphi = 30^o$ tilt control (the best-performing case for $C_T' = 3$); from the figure it is seen that the maximum power is produced in the H=500m

ABL regardless of the $C'_T$ value. It is also confirmed that, for all the considered H, a substantial power increase is obtained by increasing $C'_T$ from 1.5 to 2.25 (a 0.75 increase) but that a further increase of the same amount from 2.25 to 3 results in smaller additional power gains.

Finally, in Fig. B2 is shown the combined effect of changing $C'_T$ and the relative turbine size on the power produced with $\varphi = 30^o$ tilt control in the H=500m boundary layer; from this figure it is seen that the maximum power is produced when the ratio of the turbines diameter to the ABL momentum thickness is $D/\delta_2 \approx 3.6$ for all the considered $C'_T$ values and that higher power gain enhancements are obtained when $C'_T$ is increased from 1.5 to 2.25 than when it is increased from 2.25 to 3.

These additional results therefore show that: (a) non negligible power gains are obtained by tilt control operating the turbines at the usual induction ($C'_T = 1.5$), (b) significant power gain enhancements are already obtained by operating tilted turbines at $C'_T = 2.25$ instead of the nearly upper-bound value $C'_T = 3$ and (c) the optimal $D/\delta_2$ ratio where tilt-control power gains are maximized is not sensitive to the $C'_T$ operational value used in tilted turbines.

*Author contributions.* C.C. designed the research, performed the numerical simulations, analyzed the data, made the figures, planned, wrote and revised the paper.

*Competing interests.* The author declares that there is no conflict of interest.

*Data availability.* Data can be obtained from the author upon request.

*Acknowledgements.* I gratefully acknowledge the use of the Simulator for On/Offshore Wind Farm Applications (SOWFA) developed at NREL (Churchfield et al., 2012) based on the OpenFOAM finite volume framework.(Jasak, 2009; OpenCFD, 2011)

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

| Case | $H$ | $L_z$ | $\delta_2$ | $\Delta z_{CI}$ | $d\theta/dz_{CI}$ |
|------|-----|-------|-----------|-----------------|-------------------|
| H750 | 750m | 1000m | 54m | 100m | 0.05K/m |
| H500 | 500m | 700m | 50m | 100m | 0.05K/m |
| H350 | 350m | 600m | 47m | 50m | 0.10K/m |

**Table 1.** Characteristics of the considered atmospheric boundary layers and of their capping inversions. $L_z$ denotes the height of the simulation domain and $\delta_2$ the (streamwise) momentum thickness.

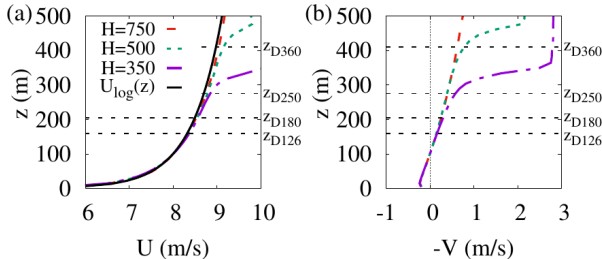

**Figure 1.** Mean-wind profiles obtained in the precursor simulations along the $x$ axis (panel $a$) and the $y$ axis (panel $b$) for the simulations with a capping inversion at $H = 750m$, $H = 500m$ and $H = 350m$. In all cases the flow is forced so as to have $U(z = 100m) = 8m/s$, $V(z = 100m) = 0$. The logarithmic velocity profile $U = (u_*/\kappa) \ln(z/z_0)$ is also reported for comparison, where $u_*$ is the mean friction velocity and $\kappa = 0.38$ is the von Karman constant. The heights corresponding to the maximum vertical extent ($z_h + D/2$) for the turbines considered in the following (with $D = 126m$, $D = 180m$, $D = 250m$ and $D = 360m$) are also reported for future reference.

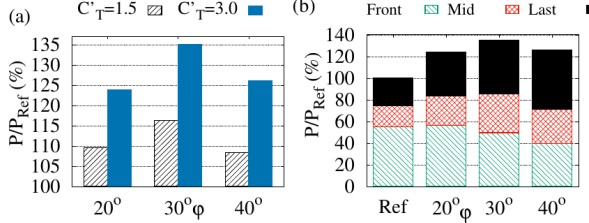

**Figure 2.** Influence of the tilt angle $\varphi$ on the average power extracted by the arrays of $D = 126m$ turbines in the ABL with $H = 750m$. ($a$) dependence on $\varphi$ of the ratio $P/P_{Ref}$ of the produced power $P$ to the $P_{Ref}$ power produced for the reference case when tilted-turbines are operated at the reference $C'_T = 1.5$ (hatched, black) or at the higher $C'_T = 3$ (solid, blue). ($b$) Decomposition of $P/P_{Ref}$ into the contributions of the upwind tilted (green, hatched, bottom), middle tilted (red, cross-hatched, middle) and downwind not tilted (black, solid, top) turbine rows when tilted-rotor turbines are operated at $C'_T = 3$.

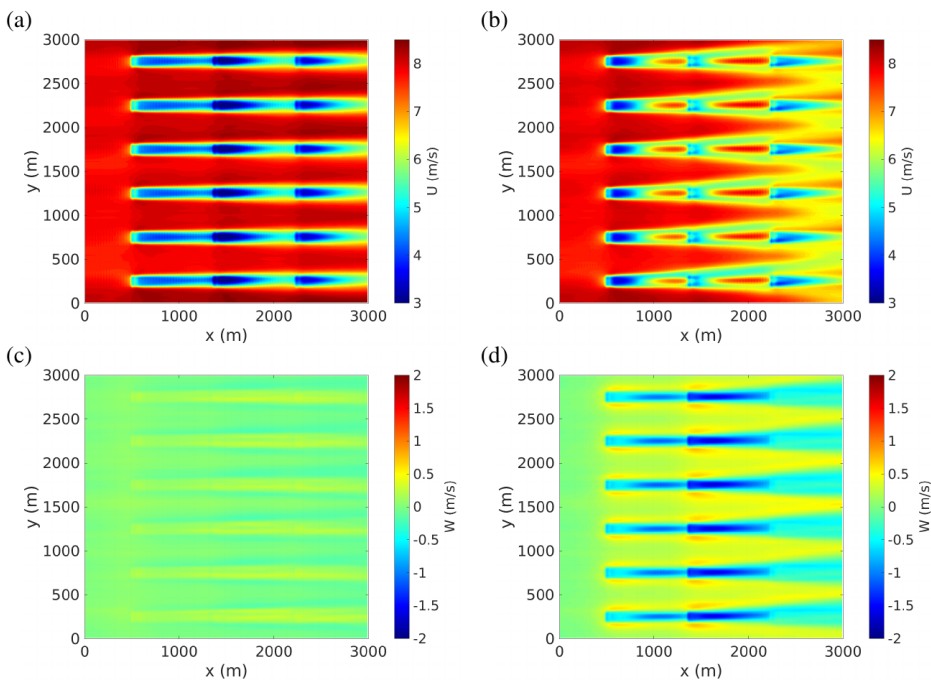

**Figure 3.** Time-averaged streamwise velocity fields (top panels $a$, $b$) and vertical velocity fields (bottom panels $c$, $d$) in the horizontal plane at hub height ($z_h = 89m$) for the simulations of the $D = 126m$ wind turbine arrays in the $H = 750m$ ABL. The incoming mean wind is from the west (left to right). Panels on the left ($a$ and $c$) pertain to the reference case while right panels ($b$ and $d$) to the case where rotors of the upwind and middle arrays are tilted by $\varphi = 30^o$ and operated at $C'_T = 3$.

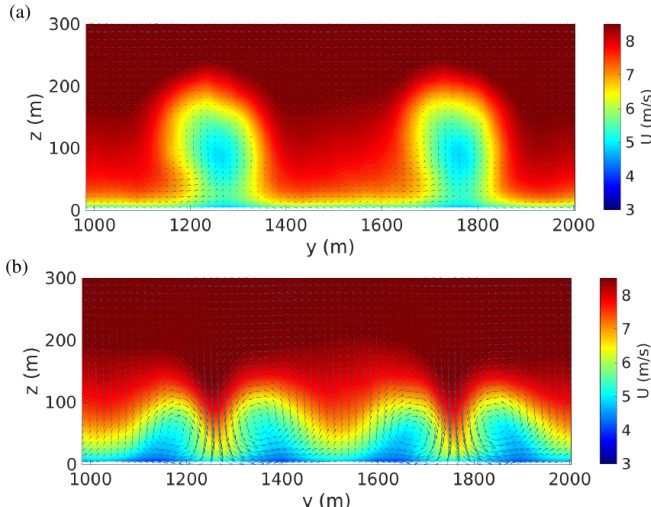

**Figure 4.** Time-averaged streamwise (color-scale) and cross-stream (arrows) velocity fields in the cross-stream plane at $x = 2000m$ in the region corresponding to the wake of the two central (in the spanwise direction) turbines of the middle turbine row (same case as in Fig. 3). Panel $(a)$: the reference case where two (low-speed) wakes are clearly visible. Panel $(b)$: the controlled case with $\varphi = 30^o$ where high-speed streaks have replaced the wakes which have been displaced towards the ground and then to the sides.

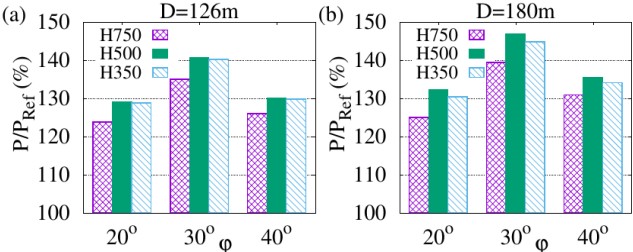

**Figure 5.** Dependence of power gains on tilt angles $\varphi$ in the ABLs with $H = 750m$, $H = 500m$ and $H = 350m$. Panel $(a)$: $D = 126m$ turbines. Panel $(b)$: $D = 180m$ turbines. Tilted-rotor turbines are operated at $C'_T = 3$.

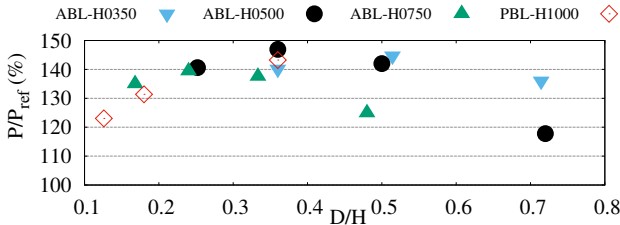

**Figure 6.** Power gains versus turbine diameter to boundary layer height ratio $D/H$ for all considered $D - H$ combinations for the controlled cases with rotors tilted by $\varphi = 30^o$ and operated at $C'_T = 3$. Data from C2020 obtained in the PBL are also reported for comparison.

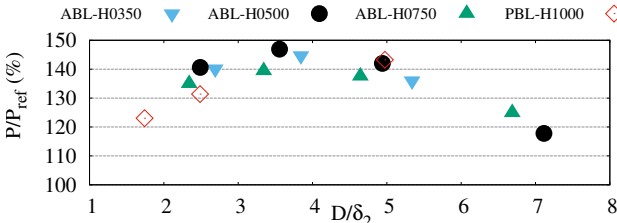

**Figure 7.** Power gains versus the ratio $D/\delta_2$ of the turbine diameter to the boundary layer momentum thickness. Same data as in Fig. 6: turbines with rotor tilted by $\varphi = 30^o$ operated at $C'_T = 3$ for all considered $D - H$ combinations with additional data from C2020 (PBL) also reported for comparison.

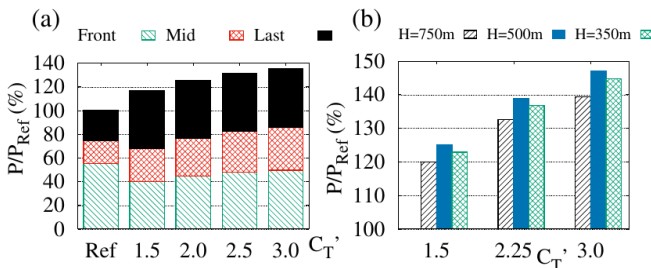

**Figure B1.** Influence of $C_T'$ on the average power extracted with $\varphi = 30^o$ tilt control: $(a)$ Power produced D=126m turbines in the H=750m ABL decomposed into front, middle and last row contributions (the baseline, no-tilt $C_T' = 1.5$, case "Ref" is also displayed for comparison); $(b)$ Power produced by D=180m turbines in the three considered ABLs (H=750m, H=500m, H=350m).

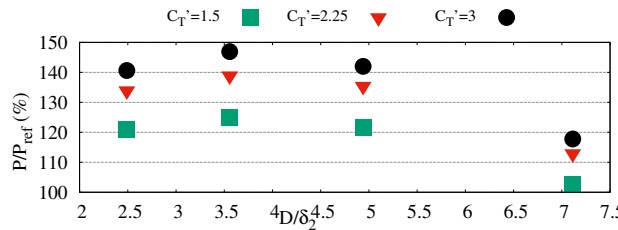

**Figure B2.** Power gains versus the ratio $D/\delta_2$ of the turbine diameter to the boundary layer momentum thickness for turbines with control rotors tilted by $\varphi = 30^o$ operated at selected $C_T'$ in the H=500m ABL.

## Appendix A

## Appendix B

465

