# Peer review of "Evaluation of tilt control for wind-turbine arrays in the atmospheric boundary layer"

_Wind Energy Science, 2020_

## Referee Comment (RC1) · Anonymous Referee #1 · 28 Nov 2020

The paper uses Large Eddy Simulations to study the effect of rotor diameter, boundary layer height, and wind veer on power gains from tilt-misalignment in a wind farm. Several interesting insights are gained, indicating the relation between streak amplification in the boundary layer, and tilt-induced power gains. The paper is well written.

The reviewer has some minor comments/ questions:

- Line 96: The actuator disk models are simulated with a constant ratio between thrust coefficient and power coefficient. When the author choses to run wind turbines at a higher thrust coefficient, how is the power coefficient modeled? If it is modeled with the linear dependence on thrust coefficient as described in the text, the power coefficient will be unrealistically high for the case of $C_T'=3$? It would be helpful if the author can explain in the text how the power coefficient is modeled when the thrust coefficient is

increased past CT'=2, and how a higher thrust coefficient can be practically applied in a real wind farm.

- If the power coefficient indeed increases to an unrealistically high value for CT'=3, it would mean that the power of the first row of turbines is overestimated? Maybe the author can add a plot of average row power also for the case of CT'=1.5?

- One focus of this paper is 'quantifying' or 'estimating' power gains from tilt misalignment. (see line 62) However, Large Eddy Simulations are not perfect, as small scale turbulence is missing. Subgrid scale modeling can have an effect on the turbulent diffusion of the counter rotating vortex pairs, leading to an over-estimate of the downstream dominance of the counter rotating vortex pair. Furthermore, the wind turbines are modeled by actuator disk models. It would be helpful for the reader if the author gives a brief discussion on the limitations of this study.

- Line 104: 'spanwise turbine spacing $\lambda y = 4D$': It is mentioned later in the text, but it would be helpful to mention here the typical spanwise spacing for 'streak generators' as described non-dimensionally in the respective papers, instead of converted into wind turbine diameters for this specific case.

- Wind veer is relatively limited in the considered boundary layer conditions. Does the author expect a bigger impact on power improvement from tilt when wind veer would be stronger?

---

## Referee Comment (RC2) · Anonymous Referee #2 · 8 Jan 2021

The author investigates the impact of rotor tilt angles and thrust coefficients on power gains of groups of turbines across varying atmospheric boundary layer heights. Additionally, the influence of rotor diameter is examined on performance gains and streak amplification across the various conditions. The performance gains found are quite significant, although they are only determined for wind-aligned operation. The paper is well written and thorough in its explanation and analyses. The author has a few minor comments/suggestions:

1) The author examines a range of positive tilt angles including 20, 30, and 40 degrees, finding across all the conditions that a tilt angle of 30 degrees gives the greatest increase in power production. The author also states that these tilt angles would best be accomplished with downwind rotors/blades. However, a tilt angle of 30 degrees

seems significantly larger than what is currently reasonable with turbine designs. The reviewer feels the reader would benefit from some discussion of the practicality of tilt angles in this range to help ground the results.

2) Along the lines of comment 1, including some discussion of the potential impact on turbine loading would be useful to the reader as well.

3) The flow diagrams in figures 3 and 4 are very useful to the reader in order to visualize the benefits of using positive tilt angles to deflect the wakes and draw higher velocity flows downwards for the downwind turbine. While the reviewer can understand why the author may have only included flow diagrams for once case in order to keep the main body of the paper concise, it could be helpful/interesting to the reader to include flow diagrams of some of the other cases in the appendices. Unless of course the flow is not significantly different, in which the author should then state that in the manuscript.

4) In the conclusion, the author acknowledges that more work is to be done to determine the gains that would be possible across a typical wind rose. The reviewer believes the paper would be strengthened by including discussion on what the results may look like in a partially waked case, as the high velocity streaks would not be as well aligned with the downwind turbines, and could even cause undesirable loads across the rotor.

---

## Author Comment (AC1) · 30 Jan 2021

Please see the supplement zip file

Please also note the supplement to this comment:
https://wes.copernicus.org/preprints/wes-2020-106/wes-2020-106-AC1-supplement.zip

---

## Author Comment (AC2) · 30 Jan 2021

Please see the supplement zip file

Please also note the supplement to this comment:
https://wes.copernicus.org/preprints/wes-2020-106/wes-2020-106-AC2-
supplement.zip

---

## Author Response (AR1)

Carlo Cossu
*Directeur de recherche CNRS*
*Head of the Urban & Coastal Atmospheric Dynamics Group*
LHEEA, CNRS - Centrale Nantes
1 rue de la Noë, 44300 Nantes, France
☎ +33(0)240376824, ✉ carlo.cossu@ec-nantes.fr

[Figure]

[Figure]

Prof. R.B. Cal
Associate Editor
*Wind Energy Science*

Nantes, 30 January 2021

Dear Pr. Cal,

I thank you and the reviewers for having examined the manuscript entitled "Evaluation of tilt control for wind-turbine arrays in the atmospheric boundary layer".

Both reviewers expressed a globally positive opinion of the manuscript but also had a number of comments and suggestions that have been addressed in the revised manuscript and in the reply to each reviewer. As you can read in the posted author's comments, I have followed most of their suggestions that have helped to improve the quality of the manuscript that I'm resubmitting for publication in *Wind Energy Science*.

During the revision process I have also realized that there were inconsistencies in the roughness length used in the simulations (they were different in the precursor simulations and in the simulations in the presence of turbines). I have rerun all the simulations with consistent values and updated the paper figures accordingly. Luckily, the results are not significantly changed.

All the modifications can be tracked in the highlighted  revised version of the manuscript which has also been posted (red = removed, blue = added or modified).  The main modifications are the following ones:
- All the figures and tables have been updated with the new simulations.
- Additional simulation have been performed to further analyze power gains that can be obtained with values of $C'_T$ smaller than 3. These additional results are presented and discussed in the newly-added Appendix B and are mentioned in the main text when appropriate.
- The need for a detailed structural load analysis is further emphasized in the conclusions.

I hope that you and the reviewers will find this revised version suitable for publication.

Yours sincerely,

**Comments on the review by Reviewer 1 of "Evaluation of tilt control for wind-turbine arrays in the atmospheric boundary layer"**

Carlo Cossu

Laboratoire d'Hydrodynamique Énergetique et Environnement Atmosphèrique (LHEEA)

CNRS - Centrale Nantes, 1 rue de la Noë 44300 Nantes, France

January 30, 2021

I thank Reviewer 1 for the many constructive comments and suggestions which have helped to improve the manuscript.

During the revision process I became aware that most of the simulations had problems in the roughness lengths values selected by the input files (in particular the roughness lengths of the precursor simulation where different from those of the simulations with the turbines). All the simulations have therefore been repeated with consistent correct values ($z_0 = 0.001$) and the manuscript has been modified accordingly. The main results are not changed, so that the conclusions of the study are not affected by these updated results (but, where appropriate, some quantitative values have been updated in the revised manuscript, as can be seen in the highlighted copy of the manuscript).

Following the referees comments and suggestions, the manuscript has undergone a non-negligible revision, where the main modifications are the following:

- All figures and tables have been updated with the results from the new simulations (with the correct consistent value of $z_0$). Changes resulting from these new simulations are updated in the revised manuscript.

- Additional simulation have been performed to further analyze power gains that can be obtained with $C_T' < 3$ values. These additional results are presented and discussed in the newly-added Appendix B and are mentioned in the main text when appropriate.

- The need for a detailed structural load analysis is further emphasized in the conclusions.

Each issue raised by a specific comment in the report is addressed in detail below. Modifications of the manuscript can be tracked in the highlighted version of the revised article (red = removed, blue = added or modified).
———————

*The paper uses Large Eddy Simulations to study the effect of rotor diameter, boundary layer height, and wind veer on power gains from tilt-misalignment in a wind farm. Several interesting insights are gained, indicating the relation between streak amplification in the boundary layer, and tilt-induced power gains. The paper is well written. The reviewer has some minor comments/ questions.*

I am glad of this positive general opinion on the manuscript. The comments/questions are addressed below.

*- Line 96: The actuator disk models are simulated with a constant ratio between thrust coefficient and power coefficient. When the author choses to run wind turbines at a higher thrust coefficient, how is the power coefficient modeled? If it is modeled with the linear dependence on thrust coefficient as described in the text, the power coefficient will be*

[Figure]

Figure R1.1 Dependence of local thrust (panel $a$) and power (panel $b$) coefficients $C'_T$, $C'_P$ of the rotor-based power coefficient on the rotor-collective blade-pitch angle ($\beta$) computed from simulation data on SOWFA's original ADM model of the NREL-5MW turbine, based on the blade element method. Power coefficient predicted from the thrust coefficient as $\chi C'_T$ (with $\chi = 0.9$) are also reported for comparison in panel $b$. The results shown pertain to time-averaged $C'_T$ and $C'_P$ values computed for each turbine of the upwind (front) row of the array in the reference case (no tilt no pitch) and for the $\varphi = 30^o$ tilted case for a range of $\beta$ values.

*unrealistically high for the case of CT'=3? It would be helpful if the author can explain in the text how the power coefficient is modeled when the thrust coefficient is increased past CT'=2, and how a higher thrust coefficient can be practically applied in a real wind farm.*

There are many questions nested in this point. Let me address them separately:

−The power coefficient is always (even when $C'_T > 2$) modeled as $C'_P = \chi C'_T$ with $\chi = 0.9$, as described in the text.

− The $C'_P = \chi C'_T$ modeling is accurate according to Munters & Meyers (2017, cited in the manuscript) who test, by LES, the same simplified actuator disk model used in the present study against theoretical predictions (see their Appendix A). I have further checked this by performing simulations with SOWFA's original actuator disk model, which is based on the blade-elements method, using the rotor-collective blade-pitch angle $\beta$ to change the turbine load and then computing $C'_T$ and $C'_P$ from simulation outputs. The computed $C'_T(\beta)$ and $C'_P(\beta)$ data are shown in Figure R1.1 for the case of tilt control in a two-rows array of NREL 5-MW wind turbines with the same spacing ($4D - 7D$) and flow parameters as in the present study for the $H = 750\,m$ ABL (the $C'_T$ data are reproduced from the study "Wake redirection at higher axial induction" which is currently under review in Wind En. Sci., DOI: 10.5194/wes-2020-111, cited as Cossu, 2021 in the manuscript). From this figure it can be verified that the relation $C'_P = \chi C'_T$ (with $\chi = 0.9$) is a good approximation of the data even for $C'_T > 2$.

−For $C'_T = 3$, the power coefficient is $C'_P = 2.7$ (as shown above) which is not unrealistic. As mentioned, the additional simulations I have performed by means of SOWFA's model of the NREL-5MW turbine, show that the large $C'_T = 3$ value can be obtained with a rotor-collective blade-pitch angle $\beta = -5^o$. Furthermore, many other studies use such large values of $C'_T$ in numerical simulations of wind farm control. For instance, Goit & Meyers (2015, cited in the manuscript) show that large $C'_T$ values (up to $C'_T = 4$) can be accessed by modified NREL-5MW models by increasing the wind-tip speed ratio or/and the blades chord length (see their Appendix A). Munters in his PhD dissertation (2017, cited in the manuscript) shows that such large values of $C'_T$ can be obtained by simply increasing the tip speed ratio provided that the mean wind speed is not too large (see his Figure 2.4) and Munters & Meyers (2017, cited in the manuscript, and a few other papers), indeed enforce a maximum value $C'_T = 3$ in their computations.

− In the revised manuscript it is now mentioned (in the new Appendix B) that high $C'_T$ values could be obtained by acting on the rotor-collective blade-pitch angle and/or increasing the tip speed ratio of typical existing wind turbines or, in the design phase, by increasing chords lengths (lines 310-312). It is also mentioned that: (a) one does not necessarily need to enforce values as high as $C'_T = 3$ because significant power gains can already be obtained for lower values of $C'_T$ (lines 230-232) and (b) the issue of structural loads for higher $C'_T$ high-tilt turbines has to be addressed mostly for next-generation wind turbines (lines 282-289).

[Figure]

**Figure R1.2** Influence of $C_T'$ on the average power extracted with $\varphi = 30^o$ tilt control from D=126m turbines in the H=750m ABL decomposed into front, middle and last row contributions (the baseline, no-tilt $C_T' = 1.5$, case "Ref" is also displayed for comparison).

*- If the power coefficient indeed increases to an unrealistically high value for CT'=3, it would mean that the power of the first row of turbines is overestimated? Maybe the author can add a plot of average row power also for the case of CT'=1.5?*

As discussed in the previous point, I am not convinced that $C_P'$ grows to an unrealistically high value for $C_T' = 3$. However, I understand the concern that $C_T' = 3$ might be considered unfeasible lacking additional studies on the feasibility of turbines capable of high positive tilt angles combined with overloading. I have therefore added a plot of average row power not only for $C_T' = 1.5$, as suggested, but also for the intermediate values $C_T' = 2$ and $C_T' = 2.5$ in the new Fig.B1$a$, reproduced here in Fig.R1.2. From this figure it can be seen that significant power gains can be obtained also at intermediate $C_T'$ values and that only (roughly) half of the power gains enhancements obtained by increasing $C_T'$ do come from first-row turbines.

For the same reason, I have added further additional results in Appendix B showing that: (a) non-negligible power gains are obtained by tilt control operating the turbines at the reference $C_T' = 1.5$ value, (b) significant power gain enhancements are already obtained by operating tilted turbines at $C_T' = 2.25$ (instead of the almost borderline value $C_T' = 3$) and (c) the optimal $D/\delta_2$ ratio where tilt-control power gains are maximized is not sensitive to the $C_T'$ operational value used in tilted turbines.

*- One focus of this paper is quantifying' or estimating' power gains from tilt misalignment. (see line 62) However, Large Eddy Simulations are not perfect, as small scale turbulence is missing. Subgrid scale modeling can have an effect on the turbulent diffusion of the counter rotating vortex pairs, leading to an over-estimate of the downstream dominance of the counter rotating vortex pair. Furthermore, the wind turbines are modeled by actuator disk models. It would be helpful for the reader if the author gives a brief discussion on the limitations of this study.*

I agree that some of the potential limitations of the study were not mentioned or discussed enough in the original manuscript. That further investigations removing these limitations are needed is mentioned in the conclusions of the revised manuscript (lines 263-276).

*- Line 104: 'spanwise turbine spacing $\lambda_y = 4D$': It is mentioned later in the text, but it would be helpful to mention here the typical spanwise spacing for 'streak generators' as described non-dimensionally in the respective papers, instead of converted into wind turbine diameters for this specific case.*

The streaks generators used in previous flat-plate investigations and mentioned in the manuscript were wall-mounted cylinders of diameter $d$, spanwise spacing $\Delta z$ and height $k$ (see Fig. R1.3). During long preliminary work to the 2004 paper it was found that streaks with most of the energy in the first spanwise harmonic (i.e. with the streamwise velocity profile approaching a pure sinus in the spanwise direction) could be obtained when $d/\Delta z \approx 1/4$.

Considering wind turbines, the formation mechanism of the streamwise vortices (by lift induced by the tilted rotors) is different from the one of the roughness elements (which is probably a mix of horseshoe vortex formation and lift-induced mechanisms) but the initial spanwise spacing of the two counter-rotating vortices is roughly $D$ for the turbines (even if tilted, the rotor spanwise size is always $D$) as in was $d$ for the roughness elements (the spanwise size of the roughness elements is the cylinder diameter). This is why the same terminology and adimensionalisation is used here.

As this was probably unclear in the manuscript, the explanation has been made more precise in the revised manuscript (line 108) by mentioning that the roughness elements are circular and that D is their diame-

[Figure]

Figure R1.3 Scheme of the spanwise array of wall-mounted cylindrical roughness elements used to generate streamwise streaks in the flat-plate boundary layer (reproduced from Fig. 13 in Fransson et al. 2004, cited in the manuscript ).

ter/width.

*- Wind veer is relatively limited in the considered boundary layer conditions. Does the author expect a bigger impact on power improvement from tilt when wind veer would be stronger?*

This is a very interesting question. I would expect that a stronger wind veer would result in an increased lateral deviation of the high-speed streaks because the higher-speed fluid which is pushed down by the vortices has a stronger lateral component. This would probably require to combine tilt-control with an additional yaw adjustment able to counter this lateral displacement. This is mentioned in the conclusions of the revised manuscript (line 261).

————

I hope to have clarified the main issues raised in the report. I thank again Reviewer 1 for his/her remarks and suggestions which have helped to improve the manuscript.

**Comments on the review by Reviewer 2 of "Evaluation of tilt control for wind-turbine arrays in the atmospheric boundary layer"**

Carlo Cossu

Laboratoire d'Hydrodynamique Énergetique et Environnement Atmosphèrique (LHEEA)

CNRS - Centrale Nantes, 1 rue de la Noë 44300 Nantes, France

January 30, 2021

I thank Reviewer 2 for his/her comments and suggestions which have helped to improve the manuscript.

During the revision process I became aware that most of the simulations had problems in the roughness lengths values selected by the input files (in particular the roughness lengths of the precursor simulation where different from those of the simulations with the turbines). All the simulations have therefore been repeated with consistent correct values ($z_0 = 0.001$) and the manuscript has been modified accordingly. The main results are not changed, so that the conclusions of the study are not affected by these updated results (but, where appropriate, some quantitative values have been updated in the revised manuscript, as can be seen in the highlighted copy of the manuscript).

Following the referees comments and suggestions, the manuscript has undergone a non-negligible revision, where the main modifications are the following:

- All figures and tables have been updated with the results from the new simulations (with the correct consistent value of $z_0$). Changes resulting from these new simulations are updated in the revised manuscript.

- Additional simulation have been performed to further analyze power gains that can be obtained with $C'_T < 3$ values. These additional results are presented and discussed in the newly-added Appendix B and are mentioned in the main text when appropriate.

- The need for a detailed structural load analysis is further emphasized in the conclusions.

Each issue raised by a specific comment in the report is addressed in detail below. Modifications of the manuscript can be tracked in the highlighted version of the revised article (red = removed, blue = added or modified).
————————

*The author investigates the impact of rotor tilt angles and thrust coefficients on power gains of groups of turbines across varying atmospheric boundary layer heights. Additionally, the influence of rotor diameter is examined on performance gains and streak amplification across the various conditions. The performance gains found are quite significant, although they are only determined for wind-aligned operation. The paper is well written and thorough in its explanation and analyses.*

I am glad of this positive general opinion on the manuscript.

*1) The author examines a range of positive tilt angles including 20, 30, and 40 degrees, finding across all the conditions that a tilt angle of 30 degrees gives the greatest increase in power production. The author also states that these tilt angles would best be accomplished with downwind rotors/blades. However, a tilt angle of 30 degrees seems*

*significantly larger than what is currently reasonable with turbine designs. The reviewer feels the reader would benefit from some discussion of the practicality of tilt angles in this range to help ground the results.*

Clearly, it is not possible/practical to implement the type of tilt control analyzed in this study for current-generation wind turbines, especially with the computed optimal tilt angles. However, it might be possible to implement it in future generations of wind turbines if positive tilt-capabilities are required in the design phase. The revised manuscript has been to make this clear in both the introduction (lines 29-34) and the conclusions (lines 280-289).

*2) Along the lines of comment 1, including some discussion of the potential impact on turbine loading would be useful to the reader as well.*

From a speculative point of view, I do not see critical loading problems in the significant-tilt high-$C'_T$ regime because a significant tilt greatly reduces the $u_d$ wind component normal to the rotor and therefore the thrust force which scales like $u_n^2$; in this context an increase of $C'_T$ would, probably only partially, compensate the reduction of the thrust force (which scales linearly in $C'_T$). An issue could come from the blade-bending moment generated by the gravity force which would develop a component normal to the rotor and directed towards the ground when the rotor is tilted; however, the tilt would also induce a vertical component in the aerodynamics forces which is directed in the opposite vertical direction as the generation of the streamwise vortices is due to a positive lift force on the rotor; in this context, increasing $C'_T$ would probably be beneficial to counteract the bending moments induced by gravity forces.

I have refrained from including the above discussion in the manuscript because it would be too speculative and potentially misleading if unsupported by data. The problem is that providing such data, besides being outside the scope of this study, would be problematic. Indeed, I could have run a few simulations using the FAST model in SOWFA which would provide load data for the NREL-5MW turbine but the relevance of these hypothetical results would be questionable because (a) this turbine is the smallest one considered in this study and it has not the optimal size for any of the considered ABLs and (b) the NREL5-MW represents current-generation wind turbines which have not been designed to be tilted by large angles, or even with the required positive tilt angles as they have upwind-facing rotors. It would make much more sense to compute the loading of turbines such as the SUMR-13MW model considered by Bay et al. (Flow Control Leveraging Downwind Rotors for Improved Wind Power Plant Operation, 2019 American Control Conference, Philadelphia, PA, USA, July 10-12, 2019) but I have no access to a FAST model for it that I could use in SOWFA (and creating it would go much beyond the scope of this study) and, probably, even if I had it, it should probably go through an additional design phase to optimize the blades for high-tilt operation.

In the revised manuscript I therefore mention (lines 277-289) that additional studies are needed where structural loads are computed for tilted turbines operated at high $C'_T$, especially for next-generation wind turbines.

*3) The flow diagrams in figures 3 and 4 are very useful to the reader in order to visualize the benefits of using positive tilt angles to deflect the wakes and draw higher velocity flows downwards for the downwind turbine. While the reviewer can understand why the author may have only included flow diagrams for once case in order to keep the main body of the paper concise, it could be helpful/interesting to the reader to include flow diagrams of some of the other cases in the appendices. Unless of course the flow is not significantly different, in which the author should then state that in the manuscript.*

I appreciate that the reviewer understands the effort put in trying to keep the paper short and readable.

I reproduce a sample of flow diagrams in the three figures below. From Fig. R.2.1, showing the case of the D=180m wind turbines in the three considered ABLs with H=750m, H=500m and H=350m respectively, it is seen that the flows are extremely similar in the three ABLs and very similar to those shown in Fig. 3 of the manuscript. From Figs. R.2.2 and R.2.3, showing the case of the four considered turbine diameters (D=126m, D=180m, D=250m, D=360m) in the H=500m ABL, it is seen that all the flows are similar except for the southerly wind veer effect which begins to become significant for the largest considered value of the ratio D/H=0.72 corresponding to the D=360m turbine in the H=500m ABL. This is now explicitly stated in the revised manuscript (lines 189-192).

For the sake of focus and conciseness of the manuscript I prefer not to reproduce in it, in an additional appendix, these extra fields which do not add much to the discussion (I have already added an appendix presenting additional results at intermediate $C'_T$ values).

[Figure]

Fig. R1.1 Time-averaged vertical (top row) and streamwise (bottom row) velocity fields in the horizontal plane at hub height for the simulations of the D=180 turbines in the H=750m (panels $a$ and $d$), H=500m (panels $b$ and $e$) and H=350m ABLs (panels $a$ and $d$) with control turbines tilted by $\varphi = 30^o$ and operated at $C'_T = 3$.

*4) In the conclusion, the author acknowledges that more work is to be done to determine the gains that would be possible across a typical wind rose. The reviewer believes the paper would be strengthened by including discussion on what the results may look like in a partially waked case, as the high velocity streaks would not be as well aligned with the downwind turbines, and could even cause undesirable loads across the rotor.*

This is a very interesting remark. Probably the best fix for a partially-"streaked" case would be to add a bit of yaw control to steer the high-speed streak towards the downwind rotor and keep it "totally streaked". This is now mentioned in the conclusions of the revised manuscript (lines 257-261).

————

I hope to have clarified the main issues raised in the report. I thank again Reviewer 2 for his/her comments and suggestions which have helped to improve the manuscript.

[Figure]

Fig.R2.2 Time-averaged vertical velocity fields in the horizontal plane at hub height for the simulations in the H=500m ABL of the arrays of (a) D=126m, (b) D=180, (c) D=250m, (d) 360m wind turbines with control turbines tilted by $\varphi = 30^o$ and operated at $C'_T = 3$.

[Figure]

Fig.R2.3 Time-averaged streamwise velocity fields in the horizontal plane at hub height for the simulations in the H=500m ABL of the arrays of (a) D=126m, (b) D=180, (c) D=250m, (d) 360m wind turbines with control turbines tilted by $\varphi = 30^o$ and operated at $C'_T = 3$.